# Improve Generalization and Robustness of Neural Networks via Weight Scale Shifting Invariant Regularizations

**Ziquan Liu** [1]  **Yufei Cui** [1]  **Antoni B. Chan** [1]

## Abstract

Using weight decay to penalize the L2 norms of weights in neural networks has been a standard training practice to regularize the complexity of networks. In this paper, we show that a family of regularizers, including weight decay, is ineffective at penalizing the intrinsic norms of weights for networks with positively homogeneous activation functions, such as linear, ReLU and max-pooling functions. As a result of homogeneity, functions specified by the networks are invariant to the shifting of weight scales between layers. The ineffective regularizers are sensitive to such shifting and thus poorly regularize the model capacity, leading to overfitting. To address this shortcoming, we propose an improved regularizer that is invariant to weight scale shifting and thus effectively constrains the intrinsic norm of a neural network. The derived regularizer is an upper bound for the input gradient of the network so minimizing the improved regularizer also benefits the adversarial robustness. We demonstrate the efficacy of our proposed regularizer on various datasets and neural network architectures at improving generalization and adversarial robustness.

## 1. Introduction

Weight decay (Krogh & Hertz, 1992) is a common regularization in training deep neural networks, i.e., to regularize the sum of squared L2 norms of the weights. However, when the network has positively homogeneous activation functions (PHAFs), i.e. $\phi(ax) = a\phi(x)$ for $a > 0$, the scale of weights can be shifted between layers without changing the input-output function specified by the network. Consider a 2-layer neural network with input $\boldsymbol{x} \in \mathbb{R}^d$ and weights $\boldsymbol{W}_1 \in \mathbb{R}^{d \times h}$ and $\boldsymbol{w}_2 \in \mathbb{R}^h$. Assuming the activation func-

tion $\phi$ is positively homogeneous, e.g., ReLU or linear function, we have $y = \boldsymbol{w}_2^T \phi(\boldsymbol{W}_1^T x) = a\boldsymbol{w}_2^T \phi(\frac{1}{a}\boldsymbol{W}_1^T x)$ for $a > 0$. This implies that the weight decay cannot well regularize the intrinsic norm of networks; we can change the weight decay term by shifting weight magnitudes between layers while maintaining the network function. Here, we further consider a family of regularizers, which includes weight decay, and prove that they can always be minimized by properly shifting scales without changing the network function. Since PHAFs are common in modern networks, overcoming this difficulty of weight decay, and its larger family of regularizers, is of great significance.

In this paper, we propose to regularize the intrinsic norms of the networks using an improved norm regularizer, which is invariant to shifting of weight scales between layers. Based on the homogeneity of activation functions, we extract the weight magnitudes $\eta(\boldsymbol{W}_i)$ of all layers, measured by some pre-defined norms (e.g., $l_p$ or spectral norm), resulting in a product of norms for the whole network, i.e., $\prod_{i=1}^{L} \eta(\boldsymbol{W}_i)$. Within each layer, this leaves a normalized weight $\boldsymbol{W}_i/\eta(\boldsymbol{W}_i)$ that should also be penalized to induce sparsity. Thus we have two terms in the improved regularizer: the overall weight magnitude of the network, and the complexity of each layer. We next show that besides the WEight-Scale-Shift-Invariance (WEISSI) property, our regularizer improves adversarial robustness of neural networks. Previous works (Simon-Gabriel et al., 2019; Ross & Doshi-Velez, 2018; Cisse et al., 2017) have proved that penalizing the input gradient, or similarly the Lipschitz constant of a network, increases the adversarial robustness. In this paper, we upper bound the norm of the input gradient by the weight energy term in our regularizer, which is previously shown to be related to the Lipschitz constant. By deriving the input gradient, we also find that sparsifying the activation maps has a positive effect on adversarial robustness. Finally, we empirically prove the effectiveness of WEISSI regularizers on various neural architectures and datasets.

## 2. Preliminaries

Consider a neural network $\hat{\boldsymbol{y}} = f(\boldsymbol{x}; \boldsymbol{\Theta}) : \mathbb{R}^D \mapsto \mathbb{R}^P$, where $\boldsymbol{\Theta}$ are the weights and other trainable parameters. The neural network has $L$ hidden layers and the function is

[1]Department of Computer Science, City University of Hong Kong. Correspondence to: Ziquan Liu <ziquanliu2-c@my.cityu.edu.hk>.

*Accepted by the ICML 2021 workshop on A Blessing in Disguise: The Prospects and Perils of Adversarial Machine Learning.* Copyright 2021 by the author(s).

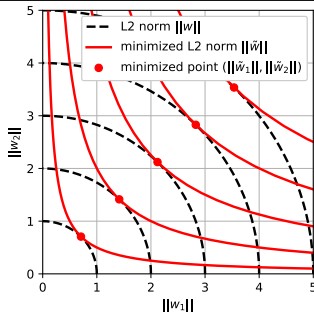

*Figure 1.* By shifting weight scales between layers, networks on the red line have the same minimized $l_2$-norm $\|\tilde{w}\|$, but have different actual $l_2$-norms $\|w\|$ (dashed lines).

recursively composed as

$$\hat{y} = W_{L+1}\phi(W_L h_{L-1} + b_L), \qquad (1)$$

$$h_l = \phi(W_l h_{l-1} + b_l), \forall l = 1, \cdots, L. \qquad (2)$$

We define the input $x$ as $h_0$ for convenience. Weight matrix $W_l = [w_{l,1}^T; \cdots; w_{l,H_l}^T] \in \mathbb{R}^{H_l \times H_{l-1}}$, $H_l$ is the size of layer $l$, and the network as $\mathcal{W} = (W_1, \cdots, W_{L+1})$. The activation functions $\phi(\cdot)$ are assumed to be positively homogeneous, i.e., $\phi(ax) = a\phi(x)$ when $a > 0$. Many common activations in modern DNNs, such as ReLU, max pooling and average pooling, are positively homogeneous. For simplicity, our analysis assumes that the output layer is linear – for networks with non-linear output layers, our analysis applies to the sub-network containing PHAFs. The network is trained by minimizing a data loss term $\mathcal{L}_{data}$ (e.g., MSE) and a regularization term $\mathcal{L}_{reg}$,

$$\Theta^* = \arg\min_{\Theta} \mathcal{L}_{data}(\mathcal{D}; \Theta) + \lambda\mathcal{L}_{reg}(\Theta), \qquad (3)$$

and $\lambda$ is a tradeoff parameter, and $\mathcal{D} = \{(x_i, y_i)\}_i$ the data.

**Notation.** We use $\|\cdot\|_p$ to denote $l_p$ norm for vectors. For matrices, $\|\cdot\|_{p,q}$ is defined as

$$\|W_l\|_{p,q} = \left(\|w_1\|_p^q + \cdots + \|w_{H_l}\|_p^q\right)^{\frac{1}{q}}. \qquad (4)$$

When $p = q$, we use $\|\cdot\|_p$ to represent matrix norm. The spectral matrix norm is denoted as $\|\cdot\|_\sigma$. Bold symbols are vectors or matrices while plain symbols represent scalars.

## 3. Weight Scale Shift and Ineffective Regularizers

Due to the positive homogeneity, a positive scalar can be pulled out of the activation function, i.e., $\phi(\tilde{W}_l h_{l-1} + \tilde{b}_l) = \gamma_l\phi(W_l h_{l-1} + b_l)$, where $\gamma_l W_l = \tilde{W}_l, \gamma_l b_l = \tilde{b}_l$. This factorization can be repeated from the first layer to the output, yielding an equivalent network $\tilde{\mathcal{W}} = (\tilde{W}_1, \cdots, \tilde{W}_{L+1})$,

$$\hat{y} = \left(\prod_{l=1}^{L+1} \gamma_l\right) \cdot \tilde{W}_{L+1}\phi(\tilde{W}_L h_{L-1} + \tilde{b}_L). \qquad (5)$$

Under the condition of the product term $\prod_{l=1}^{L+1} \gamma_l = 1$, a "new" network with different weights is obtained, but with exactly the same mapping function as before. In other words, the weight scale can be shifted between layers while keeping the network mapping function unchanged. Such

an equivalent transformation signals a problem with the commonly used weight decay regularizer.

**Weight decay.** The weight decay (WD) regularizer is the sum of squared $l_2$ norms of the weights,

$$\mathcal{L}_{wd} = \sum_{l=1}^{L+1} \|W_l\|_2^2. \qquad (6)$$

Note that in general, for two equivalent networks $\mathcal{W}, \tilde{\mathcal{W}}$,

$$\sum_{l=1}^{L+1} \|\tilde{W}_l\|_2^2 = \sum_{l=1}^{L+1} \gamma_l^2 \|W_l\|_2^2 \neq \sum_{l=1}^{L+1} \|W_l\|_2^2. \qquad (7)$$

Thus, equivalent networks (with the same generalization error) have different regularization penalties. As a result, the training with (3) will minimize the WD term by forming an equivalent network $\tilde{\mathcal{W}}$ to the current solution $\mathcal{W}$,

$$\min_{\gamma_1, \cdots, \gamma_{L+1}} \sum_{l=1}^{L+1} \|\gamma_l W_l\|_2^2, \quad \text{s.t.} \prod_{l=1}^{L+1} \gamma_l = 1. \qquad (8)$$

The minimum is found by Lagrange multipliers (see App. 1),

$$\mathcal{L}_{wd}^* = (L+1)\left(\prod_{l=1}^{L+1} \|W_l\|_2^2\right)^{\frac{1}{L+1}}, \qquad (9)$$

which occurs when the $l_2$ norms are the same for all layers,

$$\|\tilde{W}_1^*\|_2 = \cdots = \|\tilde{W}_{L+1}^*\|_2 = \left(\prod_{l=1}^{L+1} \|W_l\|_2\right)^{\frac{1}{L+1}}. \qquad (10)$$

Fig. 1 shows the equivalent-norm curves for a one hidden layer network. All networks on one red line have the same mapping function and generalization error, while their $\mathcal{L}_{wd}$ are different. A consequence of this result is that any network with large weights in one layer, resulting in large $l_2$ norm, can be converted into an equivalent network with lower $l_2$ norm. From another perspective, increasing the complexity of one layer will increase its $l_2$ norm, but this increase can be dampened by the other layers by shifting the weight scale around. In particular, increasing a layer's $\|W_l\|_2$ by a factor of 2 will only increase the overall minimized $l_2$ norm $\mathcal{L}_{wd}^*$ by an *effective* factor of $2^{\frac{1}{L+1}}$. Crucially, this effective factor approaches 1 (i.e., no penalty increase) as the network depth $L$ increases. Thus, our key observation is that the weight decay regularizer is ineffective for deep networks because the penalty of increasing model complexity is dampened, while minimizing the loss in (3), allowing the model to more easily overfit. In Appendix A.2, we give a more generalized ineffective regularization family.

## 4. WEISSI Regularizers

To overcome the problems of WD, we propose WEISSI regularizers to penalize the intrinsic norm of a DNN. The disadvantage of WD stems from the fact that shifting the weight scales between layers will change the sum of the L2 norm, allowing the regularizer to be artificially reduced. Thus, to address this problem, an effective regularizer should

(a) Standard training

| Model | Method | Clean | FGSM | PGD10 |
|-------|--------|-------|------|-------|
| MLP | WD | 96.98±0.10 | 88.05±0.52 | 87.68±0.72 |
| | WEISSI | **97.66±0.05** | **90.22±0.20** | **90.09±0.01** |
| CNN | WD | 98.73±0.06 | 92.28±0.42 | 92.22±0.50 |
| | WEISSI | **98.83±0.08** | **94.32±0.18** | **93.85±0.19** |

(b) Adversarial training

| Model | Method | Clean | FGSM | PGD10 |
|-------|--------|-------|------|-------|
| MLP | WD | 97.44±0.12 | 94.09±0.26 | 93.92±0.33 |
| | WEISSI | **98.09±0.07** | **95.32±0.10** | **95.19±0.10** |
| CNN | WD | **98.98±0.03** | 97.69±0.09 | 97.67±0.09 |
| | WEISSI | 98.97±0.05 | **97.75±0.05** | **97.74±0.05** |

*Table 1.* Accuracy of Standard and Adversarial Training on MNIST. FGSM and PGD10 use a max allowed perturbation of $\epsilon = 0.03$.

be invariant to shifting of weight magnitudes between layers. Our approach is to consider canonical forms of the DNN weights, where the scale is factored from each layer according to a normalization function $\eta(\boldsymbol{W}_l)$,

$$\hat{\boldsymbol{y}} = \left(\prod_{l=1}^{L+1} \eta(\boldsymbol{W}_l)\right) \frac{\boldsymbol{W}_{L+1}}{\eta(\boldsymbol{W}_{L+1})} \phi(\frac{\boldsymbol{W}_L}{\eta(\boldsymbol{W}_L)} \boldsymbol{h}_{L-1} + \tilde{\boldsymbol{b}}_L),$$

where possible choices of the normalizer are $\eta(\boldsymbol{W}) = \|\boldsymbol{W}\|_{p,q}$. Based on the canonical form, the new regularization term is separated into the overall weight energy over all layers, $\prod_l \eta(\boldsymbol{W}_l)$, and the complexity of individual layers, $\sum_l g(\boldsymbol{W}_l/\eta(\boldsymbol{W}_l))$, where possible choices of $g(\cdot)$ are $\|\cdot\|_{p,q}$ with different $p$ or $q$ to prevent the function from degenerating to constant 1. Note that both terms are invariant to moving weight scales between layers, c.f. standard WD. Adding monotonically increasing functions on top of the weight energy term will not affect the invariance property, such as $\prod_l \eta(\boldsymbol{W}_l)^n$ and $\sum_l \log \eta(\boldsymbol{W}_l)$. We choose two common norms, $\eta(\cdot) = \|\cdot\|_2$ and $g(\cdot) = \|\cdot\|_1$, and therefore the regularization term is

$$\lambda_e \prod_{l=1}^{L+1} \|\boldsymbol{W}_l\|_2^2 + \lambda_c \sum_{l=1}^{L+1} \|\boldsymbol{W}_l/\|\boldsymbol{W}_l\|_2\|_1, \qquad (11)$$

where first term, $\mathcal{L}_{we}$, is defined as the weight energy term, and the second term $\mathcal{L}_{wc}$ as the layer complexity term. For very deep neural networks, the weight energy term at initialization will explode to infinity, so we take the logarithm. The weight energy term is a measure of network capacity in (Neyshabur et al., 2015), supporting the efficacy of the new regularizer from the perspective of learning theory. The key difference here is that our work derives the WEISSI regularizers to effectively prevent overfitting during training, while (Neyshabur et al., 2015) gives a generalization bound based on the weight energy term in our WEISSI regularizer.

We analyze the effect of WEISSI on the optimization using the regularizer in (11) as an example. Taking the gradient of $\mathcal{L}_{we}$ with respect to the weight at $l$-th layer, we obtain $d\mathcal{L}_{we}/d\boldsymbol{W}_l = 2\boldsymbol{W}_l \prod_{j \neq l} \|\boldsymbol{W}_j\|_2^2$. In contrast to weight decay, whose gradient is $d\mathcal{L}_{wd}/d\boldsymbol{W}_l = 2\boldsymbol{W}_l$, the new regularizer has an extra scalar term, i.e., the weight energy of all other layers. Thus the gradient of $\mathcal{L}_{we}$ is very large if

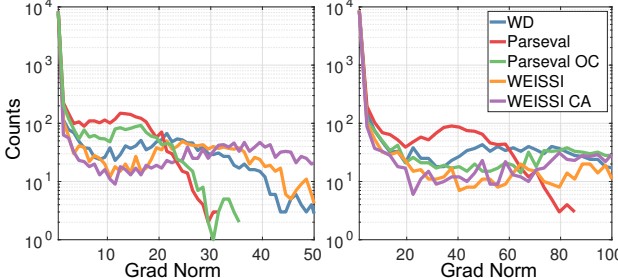

*Figure 2.* Distribution of $l_2$ norm of input gradient of WRN for (left) adversarial training and (right) standard training on CIFAR10.

the network has a large product norm, and large penalties are thus incurred if the norm is increased. For $\mathcal{L}_{wc}$, we observe that regularizing $l_1$ norm of the normalized weights induces sparse weights, which has been shown to improve adversarial robustness (Guo et al., 2018), and decrease the storage and computational time (Frankle & Carbin, 2019).

**Adversarial Robustness.** The new regularizer has an additional advantage of alleviating the vulnerability to adversarial examples because the weight energy term is an upper bound of the input gradient. Consider the gradient of the cross-entropy loss w.r.t. the input $\boldsymbol{x}$,

$$\frac{\partial \mathcal{L}_{ce}(\boldsymbol{x}, \boldsymbol{y}, \hat{\boldsymbol{y}})}{\partial \boldsymbol{x}} = \frac{\partial \hat{y}_{gt}}{\partial \boldsymbol{x}} - \sum_{j=1}^{P} p_j \frac{\partial \hat{y}_j}{\partial \boldsymbol{x}}, \qquad (12)$$

where $p_j$ and $\hat{y}_j$ are the $j$-th output of softmax function and corresponding logit, $\hat{y}_{gt}$ is the logit of the ground-truth label. For ReLU networks, we can derive an upper-bound to the input gradient norm as a function of the product norm (see Appendix A.3 for derivation),

$$\|\frac{\partial \hat{y}_j}{\partial \boldsymbol{x}}\|_2 \leq C \prod_{l=1}^{L} \|\boldsymbol{W}_l\|_2 \|\boldsymbol{w}_{L+1,j}\|_2, \qquad (13)$$

where $C$ is a constant related to sizes of layers. Hence, regularizing the product term, as WEISSI does, can help control the sensitivity of a network to adversarial perturbations. Previous works (Fazlyab et al., 2019; Guo et al., 2018) have made similar observations, but in here we derive the WEISSI regularizer from the perspective of weight scale shifting and improving adversarial robustness is a favorable property of our new regularizer. In Appendix A.4, we discuss the situation in CNNs and ResNets.

## 5. Experiments

We show the effectiveness of WEISSI in various neural network architectures on standard image recognition datasets, using weight decay and Parseval Network (Cisse et al., 2017) as baselines. More details about our experiment can be found in the Appendix B.

### 5.1. Networks without Residual Connections

We first report the effectiveness of WEISSI in regularizing DNNs without residual connections. On MNIST (LeCun et al., 1998) dataset, we use two kinds of neural architec-

| Training | Method | Clean | FGSM | | | PGD10 | |
|---|---|---|---|---|---|---|---|
| | | | $\epsilon = 0.05$ | $\epsilon = 0.03$ | $\epsilon = 0.01$ | $\epsilon = 0.03$ | $\epsilon = 0.01$ |
| Standard | WD | 90.72 | 24.43 | 26.11 | 47.36 | 13.13 | 27.67 |
| | Parseval OC | 88.72 | 6.11 | 13.33 | **48.55** | 1.46 | **34.63** |
| | WEISSI | **91.06** | **25.97** | **27.29** | 47.52 | **16.23** | 28.57 |
| Adversarial | WD | 87.64 | 23.69 | 39.68 | 69.18 | 27.67 | 65.11 |
| | Parseval OC | 84.61 | 20.24 | 36.66 | 66.93 | 26.72 | 64.06 |
| | WEISSI | **88.06** | **24.78** | **40.71** | **70.27** | **28.89** | **66.17** |

*Table 2.* Accuracy of VGG16 on CIFAR10 using different training and regularization schemes.

| Training | Method | Clean | FGSM | | | PGD10 | |
|---|---|---|---|---|---|---|---|
| | | | $\epsilon = 0.05$ | $\epsilon = 0.03$ | $\epsilon = 0.01$ | $\epsilon = 0.03$ | $\epsilon = 0.01$ |
| Standard | WD | 92.22 | 14.50 | 20.04 | 47.18 | 2.03 | 21.25 |
| | Parseval OC | **92.86** | 16.73 | 21.60 | 45.25 | 0.55 | 12.78 |
| | Parseval | 91.33 | 7.27 | 14.19 | 42.73 | 1.01 | 27.90 |
| | WEISSI | 92.45 | 23.64 | 28.50 | 50.34 | 3.03 | 16.71 |
| | WEISSI CA | 89.06 | **24.28** | **28.89** | **53.17** | **12.40** | **30.87** |
| Adversarial | WD | 89.44 | 37.57 | 51.10 | 74.23 | 41.01 | 72.56 |
| | Parseval OC | 89.39 | 32.43 | 44.07 | 74.22 | 36.84 | 73.13 |
| | Parseval | 83.82 | 24.70 | 34.96 | 65.67 | 29.43 | 64.44 |
| | WEISSI | **89.90** | 41.71 | 55.14 | **75.18** | 44.15 | **73.51** |
| | WEISSI CA | 87.69 | **42.23** | **55.22** | 72.33 | **45.39** | 70.19 |

*Table 3.* Accuracy of WRN-28-10 on CIFAR10 using different training and regularization schemes.

tures, a fully connected network (multilayer perceptron, MLP) and a convolutional neural network (CNN). The MLP has two fully connected hidden layers, and the CNN has two convolution layers. On CIFAR10 (Krizhevsky et al., 2009) dataset, we use a variant of VGG16 (Simonyan & Zisserman, 2014) as an exemplar of deep neural networks. For shallow networks on MNIST, we use $\mathcal{L}_{we}$ for the weight energy regularizer, and for VGG16 we use $\mathcal{L}_{we}^{(CNN)}$ (See Appendix A.4).

Table 1 shows the performance of WD and WEISSI on MNIST. WEISSI has a clear advantage over WD in both neural architectures and both training settings. We also observe that the variance of adversarial robustness (i.e., test accuracy on attacked images) when training with WEISSI is generally smaller than training with WD, indicating a lower sensitivity of WEISSI w.r.t. random initializations. Table 2 shows the performance of VGG16 trained on CIFAR10 using WEISSI and 2 baselines. Although Parseval OC is better at defending small perturbation ($\epsilon = 0.01$) attacks in standard training, it will deteriorate dramatically when the attack has moderate perturbation scales ($\epsilon = 0.03/05$). When using adversarial training, Parseval OC shows a large decrease in both generalization and robustness, while WEISSI remains effective and has some advantage over WD on both clean and adversarial examples. Finally, we note that adversarial training improves robustness of all architectures and regularizers when the training perturbation $\Delta = 0.01$ is closely matched to the attack perturbation $\epsilon \in \{0.01, 0.03\}$, with WEISSI showing the best performance overall.

### 5.2. Deep Residual Networks

We apply WEISSI with convex aggregation (WEISSI CA) to deep residual networks and demonstrate the advantage

of WEISSI and WEISSI CA over several baselines. We choose WRN-28-10 (Zagoruyko & Komodakis, 2016) as the architecture (28 layers and width 10). Since the WRN has many convolution layers, we use $\mathcal{L}_{we}^{(CNN)}$ as the weight energy term.

Table 3 shows the performance of WEISSI (CA) and the baselines under standard and adversarial training. For standard training, Parseval OC achieves the best generalization performance with WEISSI as the second. Nonetheless, WEISSI CA exhibits better adversarial robustness under both FGSM and PGD attacks, compared to other baselines. WEISSI also shows similar robustness as WEISSI CA to FGSM attacks, but performs poorly under PGD attacks, which shows that convex aggregation helps overall robustness. For adversarial training, Parseval performs poorly in both generalization and robustness, while Parseval OC is good against small perturbation attacks ($\epsilon = 0.01$), but again fails when the perturbation is larger. WEISSI and WEISSI CA perform similarly and both have better adversarial robustness over WD and Parseval, especially for larger $\epsilon$ attacks.

In Fig. 2, we plot the histogram of $l_2$ gradient norm of 10,000 test images to show how WEISSI (CA) changes the input gradient norms compared to several baselines. The number of gradient norms between 5.0 and 2.0 is negatively correlated with the adversarial robustness, since both WEISSI and WEISSI CA have an obvious decrease in this region compared to 3 baselines. Another finding is that Parseval network is good at suppressing relatively large input gradients, but this property does not seem to always help adversarial robustness. Thus, we speculate that the key of adversarial robustness could also be suppressing small input gradient norms.

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

## Appendix A.1 Minimized Weight Decay Regularization

We prove how to solve the optimization problem (7) in the main paper. Recall that the problem is formulated as

$$\min_{\{\gamma_l\}_{l=1}^{L+1}} \sum_{l=1}^{L+1} \|\gamma_l \boldsymbol{W}_l\|_2^2, \quad s.t. \prod_{l=1}^{L+1} \gamma_l = 1. \tag{14}$$

Introduce the Lagrange multiplier $\lambda$ and write the Lagrange function,

$$\mathcal{L}(\gamma_1, \ldots, \gamma_{L+1}, \lambda) = \sum_{l=1}^{L+1} \|\gamma_l \boldsymbol{W}_l\|_2^2 - \lambda(\prod_{l=1}^{L+1} \gamma_l - 1) \tag{15}$$

Take the gradient w.r.t. $\gamma_l$ $\forall l$ and $\lambda$, we have,

$$\frac{\partial \mathcal{L}}{\partial \gamma_l} = 2\gamma_l \|\boldsymbol{W}_l\|_2^2 - \lambda \prod_{j \neq l} \gamma_j, \quad l = 1, \ldots, L+1 \tag{16}$$

$$\frac{\partial \mathcal{L}}{\partial \lambda} = \prod_{l=1}^{L+1} \gamma_l - 1 \tag{17}$$

Let the gradient be zeros, we have $\gamma_l = (\lambda)^{1/2}/(2\|\boldsymbol{W}_l\|_2^2)^{1/2}$ and bring $\gamma_l$ back to (17), we have

$$\lambda = 2(\prod_{l=1}^{L+1} \|\boldsymbol{W}_l\|_2^2)^{1/(L+1)}. \tag{18}$$

So the optimal solution of $\gamma_l$ is $\hat{\gamma}_l = (\prod_{j=1}^{L+1} \|\boldsymbol{W}_j\|_2^2)^{\frac{1}{2(L+1)}}/\|\boldsymbol{W}_l\|_2$. It is easy to check that the second-order gradient of the objective function w.r.t. $\hat{\gamma}_l$ is greater than 0, so we obtain a minimum solution. Thus, the mimimized $l_2$ weight norm is

$$\|\hat{\boldsymbol{W}}_1\|_2^2 = \cdots = \|\hat{\boldsymbol{W}}_{L+1}\|_2^2 = (\prod_{l=1}^{L+1} \|\boldsymbol{W}_l\|_2^2)^{1/(L+1)}, \tag{19}$$

and the minimized weight decay is $(L+1)(\prod_{l=1}^{L+1} \|\boldsymbol{W}_l\|_2^2)^{1/(L+1)}$.

## Appendix A.2 An Ineffective Regularization Family

We prove Theorem 3.1 in the main paper, which states as follows,
**Theorem 3.1.** *Assume we have a regularization function* $g : \mathbb{R}^{m \times n} \mapsto \mathbb{R}, \forall m, n \in \mathbb{N}$ *and a set of real matrices* $\{\boldsymbol{W}_l\}_{l=1}^{L+1}$. *If 1)* $\exists p > 0, g(\gamma) = \gamma^p$ *and 2)* $g(\gamma \boldsymbol{W}) = g(\gamma)g(\boldsymbol{W})$, *then the following equation holds.*

$$\min_{\{\gamma_l\}_{l=1}^{L+1}} \sum_{l=1}^{L+1} g(\gamma_l \boldsymbol{W}_l) = (L+1) \sqrt[L+1]{\prod_{l=1}^{L+1} g(\boldsymbol{W}_l)}, \quad s.t. \prod_{l=1}^{L+1} \gamma_l = 1. \tag{20}$$

*proof.* Since we have $g(\gamma \boldsymbol{W}) = g(\gamma)g(\boldsymbol{W})$, the objective function can be written as

$$\min_{\{\gamma_l\}_{l=1}^{L+1}} \sum_{l=1}^{L+1} g(\gamma_l)g(\boldsymbol{W}_l). \tag{21}$$

Following the previous section, we write the Lagrange funtion,

$$\mathcal{L}(\gamma_1, \ldots, \gamma_{L+1}, \lambda) = \sum_{l=1}^{L+1} g(\gamma_l)g(\boldsymbol{W}_l) - \lambda(\prod_{l=1}^{L+1} \gamma_l - 1). \tag{22}$$

Then take the gradient of the Lagrange function,

$$\frac{\partial \mathcal{L}}{\partial \gamma_l} = g'(\gamma_l)g(\boldsymbol{W}_l) - \lambda \prod_{j \neq l} \gamma_j, \quad l = 1, \ldots, L+1 \tag{23}$$

$$\frac{\partial \mathcal{L}}{\partial \lambda} = \prod_{l=1}^{L+1} \gamma_l - 1. \tag{24}$$

Let the gradient be zero, we have

$$\gamma_l = \frac{\lambda}{g'(\gamma_l)g(\boldsymbol{W}_l)} = \frac{\lambda}{p\gamma_l^{p-1}g(\boldsymbol{W}_l)} = (\frac{\lambda}{p \cdot g(\boldsymbol{W}_l)})^{1/p}. \tag{25}$$

Then bring $\gamma_l$ back to (24), we have

$$\lambda = p(\prod_{l=1}^{L+1} g(\boldsymbol{W}_l))^{\frac{1}{L+1}}. \tag{26}$$

Thus the optimal $\gamma_l$ the minimized $g(\boldsymbol{W}_l)$ and objective function is

$$\hat{\gamma}_l = \left[\frac{\sqrt[L+1]{\prod_{j=1}^{L+1} g(\boldsymbol{W}_j)}}{g(\boldsymbol{W}_l)}\right]^{1/p}, \tag{27}$$

$$g(\hat{\boldsymbol{W}}_1) = \cdots = g(\hat{\boldsymbol{W}}_{L+1}) = \sqrt[L+1]{\prod_{l=1}^{L+1} g(\boldsymbol{W}_l)}, \tag{28}$$

$$\min_{\{\gamma_l\}_{l=1}^{L+1}} \sum_{l=1}^{L+1} g(\gamma_l \boldsymbol{W}_l) = (L+1)\sqrt[L+1]{\prod_{l=1}^{L+1} g(\boldsymbol{W}_l)}. \tag{29}$$

## Appendix A.3 WEISSI and input gradient norm

Consider the gradient of the cross-entropy loss w.r.t. the input $\boldsymbol{x}$,

$$\frac{\partial \mathcal{L}_{ce}(\boldsymbol{x}, \boldsymbol{y}, \hat{\boldsymbol{y}})}{\partial \boldsymbol{x}} = \frac{\partial \hat{y}_{gt}}{\partial \boldsymbol{x}} - \sum_{j=1}^{P} p_j \frac{\partial \hat{y}_j}{\partial \boldsymbol{x}}, \tag{30}$$

where $p_j$ and $\hat{y}_j$ are the $j$-th output of softmax function and corresponding logit, $\hat{y}_{gt}$ is the logit of the ground-truth label. Then, the input gradient is

$$\frac{\partial y_j}{\partial \boldsymbol{x}} = \prod_{l=1}^{L} (\boldsymbol{W}_l^T \boldsymbol{J}_l) \boldsymbol{w}_{L+1,j}, \tag{31}$$

$$\boldsymbol{J}_l = \text{diag}(\phi'(\boldsymbol{w}_{l,1}^T \boldsymbol{h}_{l-1} + \boldsymbol{b}_l), \cdots, \phi'(\boldsymbol{w}_{l,H_l}^T \boldsymbol{h}_{l-1} + \boldsymbol{b}_l)).$$

Taking the $l_2$ norm of input gradient, we have

$$\|\frac{\partial y_j}{\partial \boldsymbol{x}}\|_2 = \|\prod_{l=1}^{L} (\boldsymbol{W}_l^T \boldsymbol{J}_l) \boldsymbol{w}_{L+1,j}\|_2 \tag{32}$$

$$\leq \prod_{l=1}^{L} \|\boldsymbol{W}_l\|_2 \|\boldsymbol{J}_l\|_2 \|\boldsymbol{w}_{L+1,j}\|_2, \tag{33}$$

because the norm is sub-multiplicative. For ReLU network, we have $\|\boldsymbol{J}_l\|_2 \leq H_l$ (the size of layer $l$), thus obtaining an upper-bound to the input gradient norm as a function of the product norm,

$$\|\frac{\partial y_j}{\partial \boldsymbol{x}}\|_2 \leq C \prod_{l=1}^{L} \|\boldsymbol{W}_l\|_2 \|\boldsymbol{w}_{L+1,j}\|_2, \tag{34}$$

where $C$ is a constant related to sizes of layers.

## Appendix A.4 Convolution Neural Networks

A CNN uses structured convolution connections between layers, and the network function has a different expression from the fully connected case in (2). The weight scale can be shifted among layers if we regard the convolution kernel as a weight matrix. However, one problem with convolution layers is that the kernel parameters actually have more influence than weight parameters in fully connected layers, since they are actually applied $M$ times (equal to output feature map size) due to the sliding of kernel over the feature map. One solution is to multiply by $M$ in the regularizer. But the multipliers can be moved across layers due to the WEISSI property, see (11). To address this shortcoming, for deep convolution network,

we choose to sacrifice WEISSI property among all layers, and downscale the weight of the fully connected layers taking the square-root, $\mathcal{L}_{we}^{(CNN)} = \prod_{l=1}^{L} \|\boldsymbol{W}_l\|_2^2 \|\boldsymbol{W}_{L+1}\|_2$. In this way, we maintain the WEISSI property among all convolution layers and give proper penalties for their kernel parameters. From another perspective, the larger exponent on the CNN kernel weights suppresses parameters smaller than 1.0 and gives more penalty for those parameters larger than 1.0, inducing higher sparsity in the convolution kernels.

We consider the convolution layers in CNN and show that the conv layer is a special case of fully connected layer. Assume the input $\boldsymbol{z} \in \mathbb{R}^{D \times D}$ is a 2D input with one channel (for simplicity) to the convolution layer $\boldsymbol{W} \in \mathbb{R}^{d_k \times d_k \times 1 \times H}$, where $d_k$ is the kernel size and $H$ is the output channel number. We flatten the input $\boldsymbol{z}$ to a 1D vector $\tilde{z}$ with zero paddings and transform the convolution kernels so that we can write the convolution operation as $\tilde{\boldsymbol{W}}\tilde{z}$. Assuming the stride is 1 and use zero paddings to keep the input and output has the same size, the $(i, j, m)$ output element of the conv layer is

$$W_{1,1,m}z_{i-(d_k-1)/2,j-(d_k-1)/2} + W_{1,2,m}z_{i-(d_k-1)/2,j-(d_k-1)/2+1} + \cdots + W_{d_k,d_k,m}z_{i+(d_k-1)/2,j+(d_k-1)/2}. \quad (35)$$

This can be seen as the product of a sparse vector consisting of 0 and $W_{1,1,m}, \ldots, W_{d_k,d_k,m}$ and $\tilde{z}$, where the positions of kernel parameters are determined by the positions of $z_{i,j}$ so that we get the Equ. (35) from the vector product. In this way, we can have a sparse matrix $\tilde{\boldsymbol{W}}$ consisting of elements in $\boldsymbol{W}$ where each column is a sparse vector and has the same effect as the convolution operation, and the output of the conv layer can be written in the form of matrix multiplication $\tilde{\boldsymbol{W}}\tilde{z}$. Notice that the kernel parameters in the matrix are multiplied several times, which is different from parameters in fully connected layers. Specifically, the repeated computation time is the same as the output size. If we calculate the $l_2$ norm for the transformed matrix, we have $\|\tilde{\boldsymbol{W}}\|_2 = D^2 \|\boldsymbol{W}\|_2$. This suggests a more reasonable weight decay scheme: multiply the repeated computation times for convolution kernels. But the multiplication does not apply to our WEISSI regularizer, since the weight energy term is a product norm of all layers and the scale for one layer is equivalent for all layers. That is why we propose to give larger exponents for conv kernel terms to give a proper regularization for conv layers.

## Appendix A.5 Convex Aggregation in ResNet and Input Gradient Norm

We consider DNNs with residual connections (He et al., 2016), in particular Wide ResNet (WRN) (Zagoruyko & Komodakis, 2016), and obtain the same WEISSI regularizer as that of the standard network. Using our notation, a residual block in WRN is,

$$\boldsymbol{h}_1 = \boldsymbol{W}_1^{(3)}\boldsymbol{x} + \boldsymbol{W}_1^{(2)}\phi(\boldsymbol{W}_1^{(1)}\boldsymbol{x}),$$
$$\boldsymbol{h}_l = \boldsymbol{h}_{l-1} + \boldsymbol{W}_l^{(2)}\phi(\boldsymbol{W}_l^{(1)}\phi(\boldsymbol{h}_{l-1})), l = 1, \cdots, L,$$
$$\boldsymbol{y} = \boldsymbol{W}_{L+1}\phi(\boldsymbol{h}_L), \quad (36)$$

where the weights can be in the form of fully connected or convolution layers, $\boldsymbol{W}_l^{(1)} \in \mathbb{R}^{H_l^{(1)} \times H_{l-1}^{(2)}}, \boldsymbol{W}_l^{(2)} \in \mathbb{R}^{H_l^{(2)} \times H_l^{(1)}}$ and we assume in each residual block the dimensions of input and output are the same for simplicity, i.e., $H_l^{(2)} = H_{l-1}^{(2)}$. The scale in the weight matrices are pulled out of the residual blocks, i.e.,

$$\boldsymbol{h}_l = \boldsymbol{h}_{l-1} + \gamma_l^{(2)}\tilde{\boldsymbol{W}}_l^{(2)}\phi(\gamma_l^{(1)}\tilde{\boldsymbol{W}}_l^{(1)}\phi(\boldsymbol{h}_{l-1})) \quad (37)$$
$$= \gamma_l^{(2)}\gamma_l^{(1)}\{\tilde{\boldsymbol{h}}_{l-1} + \tilde{\boldsymbol{W}}_l^{(2)}\phi(\tilde{\boldsymbol{W}}_l^{(1)}\phi(\boldsymbol{h}_{l-1}))\}. \quad (38)$$

Note that we assume the shortcut connection has a scalar multiplier, e.g., as in adaptive ResNet (Zhang et al., 2018), which can be re-scaled to keep the network's function unchanged after the weight scale shifting. Repeating this transformation we obtain a product of weight scales in front of the output of the network, thus we can also use the WEISSI regularizer (11) for ResNet.

**The Effect of Convex Aggregation.** We adopt the convex aggregation strategy when summing the residual connections and feature maps (Cisse et al., 2017; Zhang et al., 2018). The residual block has extra weight scalars $0 \geq \alpha_l \geq 1$ and $0 \geq \beta_l \geq 1$, with $\alpha_l + \beta_l = 1$,

$$\boldsymbol{h}_l = \alpha_l \boldsymbol{h}_{l-1} + \beta_l \boldsymbol{W}_l^{(2)}\phi(\boldsymbol{W}_l^{(1)}\phi(\boldsymbol{h}_{l-1})), \quad (39)$$

which account for the relative importance of the shortcut $\boldsymbol{h}_{l-1}$ and feature map $\phi(\boldsymbol{W}_l^{(1)}\phi(\boldsymbol{h}_{l-1}))$. Cisse et al. (2017) explains the importance of convex aggregation by considering the Lipschitz constant of one aggregation node. Here we explain the effectiveness of convex aggregation from a global perspective, i.e., the input gradients of the whole network. We next show that convex aggregation removes an exponential term in an upper bound of the input gradient norm compared to standard ResNet, thus better constraining input gradients.

The gradient of logit $y_i$ w.r.t input $\boldsymbol{x}$ is

$$\frac{\partial y_i}{\partial \boldsymbol{x}} = \boldsymbol{W}_1^{(3)} \boldsymbol{R}_L \boldsymbol{J}_{L+1} \boldsymbol{w}_{L+1,i}, \tag{40}$$

$$\boldsymbol{R}_L = \prod_{l=1}^{L} (\alpha_l \boldsymbol{I} + \beta_l \boldsymbol{J}_l^{(1)} \boldsymbol{W}_l^{(1)^T} \boldsymbol{J}_l^{(2)} \boldsymbol{W}_l^{(2)^T}), \tag{41}$$

$$\boldsymbol{J}_l = \mathrm{diag}(\phi'(\boldsymbol{w}_{l,1}^T \boldsymbol{h}_{l-1} + \boldsymbol{b}_l), \cdots, \phi'(\boldsymbol{w}_{l,H_l}^T \boldsymbol{h}_{l-1} + \boldsymbol{b}_l)),$$

where the intermediate term $\boldsymbol{R}_L$ comprising residual blocks and aggregation scalars is our main interest. Defining the uniform upper bound for the norm of product weight terms,

$$\left\| \prod_{m \in \mathcal{S}_\beta^t} \boldsymbol{J}_m^{(1)} \boldsymbol{W}_m^{(1)^T} \boldsymbol{J}_m^{(2)} \boldsymbol{W}_m^{(2)^T} \right\|_2 \leq \sigma(\boldsymbol{\Theta}, \boldsymbol{x}), \forall t, \tag{42}$$

where $\mathcal{S}_\beta^t$ contains the $\beta_m$ weights for the $t$-th term in the product expansion of (46), we obtain an upper bound,

$$\|\boldsymbol{R}_L\|_2 \leq \left[ \prod_{l=1}^{L} (\alpha_l + \beta_l) \right] \sigma(\boldsymbol{\Theta}, \boldsymbol{x}) = \sigma(\boldsymbol{\Theta}, \boldsymbol{x}), \tag{43}$$

since $\alpha_l + \beta_l = 1$. We upper bound the input gradient as

$$\|\frac{\partial y_i}{\partial \boldsymbol{x}}\|_2 \leq H_L \|\boldsymbol{W}_1^{(3)}\|_2 \|\boldsymbol{w}_{L+1,i}\|_2 \sigma(\boldsymbol{\Theta}, \boldsymbol{x}), \tag{44}$$

where $H_L$ is from the upper bound for $\|\boldsymbol{J}_{L+1}\|_2$. In contrast, $\alpha_l = \beta_l = 1$ for standard ResNet, and thus the upper bound in (51) for these networks is $2^L \sigma(\boldsymbol{\Theta}, \boldsymbol{x})$, which contains an exponential term w.r.t network depth $L$. Thus, convex aggregation avoids the exponential dependence of the input gradient norm on network depth in residual networks.

The gradient of logit $y_i$ w.r.t input $\boldsymbol{x}$ is

$$\frac{\partial y_i}{\partial \boldsymbol{x}} = \boldsymbol{W}_1^{(3)} \boldsymbol{R}_L \boldsymbol{J}_{L+1} \boldsymbol{w}_{L+1,i}, \tag{45}$$

$$\boldsymbol{R}_L = \prod_{l=1}^{L} (\alpha_l \boldsymbol{I} + \beta_l \boldsymbol{J}_l^{(1)} \boldsymbol{W}_l^{(1)^T} \boldsymbol{J}_l^{(2)} \boldsymbol{W}_l^{(2)^T}), \tag{46}$$

where the intermediate term $\boldsymbol{R}_L$ comprising residual blocks and aggregation scalars is our main interest. We expand $\boldsymbol{R}_L$ and obtain a summation of $2^L$ summands, where the $t$th summand includes the product $\prod_{l \in \mathcal{S}_\alpha^t} \alpha_l \prod_{m \in \mathcal{S}_\beta^t} \beta_m$, and $\mathcal{S}_\alpha^t, \mathcal{S}_\beta^t$ are index sets of $\alpha_l$ and $\beta_m$ according to the expansion. In the $t$-th summand, all $\alpha_l$ and $\beta_m$ are from different multipliers/layers and $|\mathcal{S}_\alpha^t| + |\mathcal{S}_\beta^t| = L$. We write the expansion of $\boldsymbol{R}_L$ as follows,

$$\sum_{t=1}^{2^L} \prod_{l \in \mathcal{S}_\alpha^t} \alpha_l \prod_{m \in \mathcal{S}_\beta^t} \beta_m \boldsymbol{J}_m^{(1)} \boldsymbol{W}_m^{(1)^T} \boldsymbol{J}_m^{(2)} \boldsymbol{W}_m^{(2)^T}. \tag{47}$$

According to Minkowski's inequality, we have $\|\boldsymbol{R}_L\|_2$ is upper bounded by

$$\sum_{t=1}^{2^L} \prod_{l \in \mathcal{S}_\alpha^t} \alpha_l \prod_{m \in \mathcal{S}_\beta^t} \beta_m \left\| \prod_{m \in \mathcal{S}_\beta^t} \boldsymbol{J}_m^{(1)} \boldsymbol{W}_m^{(1)^T} \boldsymbol{J}_m^{(2)} \boldsymbol{W}_m^{(2)^T} \right\|_2 \tag{48}$$

The weights of a neural network cannot be infinity, thus we denote the uniform upper bound for the norm of product weight terms as $\sigma(\boldsymbol{\Theta}, \boldsymbol{x})$, i.e.,

$$\left\| \prod_{m \in \mathcal{S}_\beta^t} \boldsymbol{J}_m^{(1)} \boldsymbol{W}_m^{(1)^T} \boldsymbol{J}_m^{(2)} \boldsymbol{W}_m^{(2)^T} \right\|_2 \leq \sigma(\boldsymbol{\Theta}, \boldsymbol{x}), \forall t. \tag{49}$$

Thus, we have

$$\|\boldsymbol{R}_L\|_2 \leq \left[\sum_{t=1}^{2^L} \prod_{l \in \mathcal{S}_\alpha^t} \alpha_l \prod_{m \in \mathcal{S}_\beta^t} \beta_m\right] \sigma(\boldsymbol{\Theta}, \boldsymbol{x}) \tag{50}$$

$$= \left[\prod_{l=1}^{L}(\alpha_l + \beta_l)\right] \sigma(\boldsymbol{\Theta}, \boldsymbol{x}) = \sigma(\boldsymbol{\Theta}, \boldsymbol{x}), \tag{51}$$

since $\alpha_l + \beta_l = 1$. We upper bound the input gradient as

$$\|\frac{\partial y_i}{\partial \boldsymbol{x}}\|_2 \leq H_L \|\boldsymbol{W}_1^{(3)}\|_2 \|\boldsymbol{w}_{L+1,i}\|_2 \sigma(\boldsymbol{\Theta}, \boldsymbol{x}), \tag{52}$$

where $H_L$ is from the upper bound for $\|\boldsymbol{J}_{L+1}\|_2$.

## Appendix A.6 Related Work

We first discuss several regularizers for neural networks and theoretical work on weight norm and generalization.

**Weight decay.** Weight decay improves the generalization (Krogh & Hertz, 1992) of networks by penalizing large weight norms during training. Many modern DNNs are trained with weight decay, including VGG (Simonyan & Zisserman, 2014), ResNet (He et al., 2016) and Densenet (Huang et al., 2017). We demonstrate that weight decay is unable to control the intrinsic weight norms of a neural network with positively homogeneous activation functions, especially when the number of layers in the network becomes large. Since the neural networks that are widely used today often have ReLU activations and very deep architectures, this ineffectiveness of weight decay may cause severe problems in terms of generalization and adversarial robustness. Our WEISSI regularizers mitigate this difficulty of weight decay and thus improve the generalization. We further prove that WEISSI regularizers are preferable over weight decay in improving the adversarial robustness.

**Dropout.** Another widely used regularizer for DNNs is Dropout (Srivastava et al., 2014), which prevents co-adaptation of neurons by randomly dropping neurons during training. The implicit regularizer induced by Dropout for a single hidden-layer linear neural network is proven to constrain every hidden neuron's input and output weight norms to be equal (Mianjy et al., 2018). Such a constraint is equivalent to the square of path regularization (Neyshabur et al., 2015), which is invariant to weight scale shifting between layers. In this sense, Dropout for a single hidden-layer linear neural network has a very similar effect to that of WEISSI regularizers. WEISSI regularization extends the weight scale shift invariance property of Dropout to multi-layer neural networks. However, we emphasize that for multi-layer networks, it is complicated to analyze the implicit regularization of Dropout and its co-adaptation reduction property is generally accepted. Thus, we consider Dropout as a complementary component to WEISSI regularizer.

**Spectral Norm Regularization.** Previous works have proposed to constrain the spectral norm of a neural network to control its sensitivity to input perturbations (Yoshida & Miyato, 2017; Cisse et al., 2017). Spectral norm regularization (Yoshida & Miyato, 2017) replaces the squared L2 norm in weight decay with the squared spectral norm. However, it is still not invariant to weight scale shifting, and thus spectral norm regularization has the same drawback as weight decay. Parseval network (Cisse et al., 2017) proposes to constrain the Lipschitz constant of a neural network by introducing an orthogonal constraint on the weight matrix and a convex aggregation for residual connections. Although Parseval network and WEISSI regularizers both constrain the Lipschitz constant of neural networks, their approaches are quite different: Parseval network constrains the Lipschitz constant of each individual unit of a network to be less than 1, while WEISSI penalizes an estimated upper bound for the Lipschitz constant for a whole network. One advantage of WEISSI is that the regularizers can be easily plugged into the standard training pipeline, while Parseval network requires extra iterations to make sure the constraints are satisfied. Our experiment shows that WEISSI is more effective at improving the adversarial robustness than Parseval network. We also use the convex aggregation of Parseval network in the WEISSI regularized WideResNet (Zagoruyko & Komodakis, 2016), and achieve better performance than using Parseval network, which demonstrates the superiority of WEISSI over orthogonal weight constraints. In addition, we provide a theoretical explanation on the efficacy of convex aggregation in controlling vulnerability of networks by analyzing adversarial attacks.

**Weight Normalization.** Similar to the WEISSI formulation, weight normalization (Salimans & Kingma, 2016) disentangles the magnitude and direction of weight vectors and proposes to optimize the two components for one weight to accelerate convergence. Despite the similarity in this step, WEISSI regularization and weight normalization have different purposes: WEISSI aims to control the complexity of neural networks to mitigate overfitting, while weight normalization is dedicated

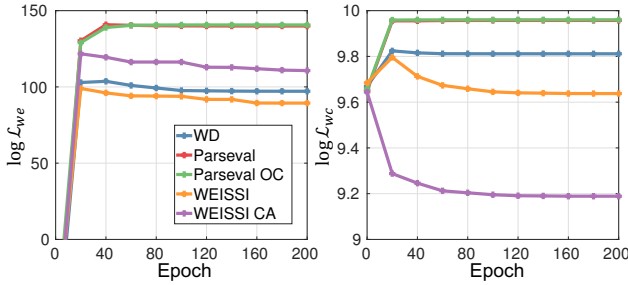

*Figure 3.* Plot of weight energy (left) and weight complexity (right) with respect to training epochs of WRN-28-10 on CIFAR10.

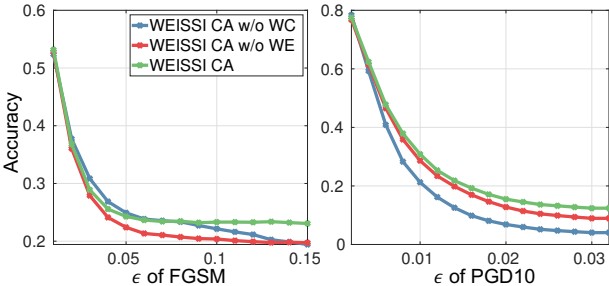

*Figure 4.* Ablation study: accuracy of WRN trained on CIFAR10 under FGSM (left) and PDG (right) attacks.

| Model | Method | Clean | FGSM | | | PGD10 | |
|---|---|---|---|---|---|---|---|
| | | | $\epsilon = 0.05$ | $\epsilon = 0.03$ | $\epsilon = 0.01$ | $\epsilon = 0.03$ | $\epsilon = 0.01$ |
| VGG | WD | 90.10 | 23.28 | 29.85 | **52.00** | 12.59 | 30.04 |
| | WEISSI | **90.64** | **33.79** | **35.98** | 50.72 | **22.31** | **33.84** |
| WRN | WD | 92.90 | 55.67 | **68.26** | **76.39** | 60.51 | 63.02 |
| | WEISSI | **93.12** | **55.98** | 67.35 | 75.64 | **60.67** | **63.67** |

*Table 4.* Accuracy of Standard Training on CIFAR10 using MMC loss.

to faster optimization. As an alternative to batch normalization (Ioffe & Szegedy, 2015), weight normalization may also hurt adversarial robustness (Xie & Yuille, 2020) since they both make the optimizer too strong, causing overfitting.

**Norm-based Generalization Error.** The proposed WEISSI regularizer is closely related to several learning theory works (Bartlett, 1998; Neyshabur et al., 2015; 2018). (Bartlett, 1998) proves that the generalization performance of a DNN depends on the magnitude of weights instead of the number of weights. (Neyshabur et al., 2015) derives a bound of the Rademacher complexity as a function of the product of weight norms, which is equivalent to the weight energy term in WEISSI regularizers. (Neyshabur et al., 2018) extends the previous work by bounding the generalization error in the PAC-Bayesian framework, where the bound is a function of the product of spectral weight norms. We note that (Neyshabur et al., 2015; 2018) both prove the invariance of a network function to weight scale shifting due to the homogeneity of ReLU activations, and use the product of some weight norms to bound generalization error. The key observation in our paper is the incapability of weight decay in penalizing the intrinsic norm of a neural network, which is not considered in the previous papers. We further propose WEISSI regularizers to more effectively control the network capacity, and thus enhance generalization accuracy and adversarial robustness. Moreover, we provide various empirical results to support our claim. In contrast, (Neyshabur et al., 2015; 2018) provide theoretical insights into the effect of the weight norm product in controlling the generalization error, without considering adversarial robustness or giving experimental evidence.

## Appendix B.1 Experiment Settings

**Dataset.** We consider the task of image recognition using deep neural networks. Our experiment is run on 2 standard image recognition datasets, i.e., MNIST (LeCun et al., 1998) and CIFAR10 (Krizhevsky et al., 2009).

|         | $\lambda_{we}$ | $\lambda_{wc}$ |         | $\lambda_{we}$ | $\lambda_{wc}$ |
|---------|---------|---------|---------|---------|---------|
| MLP     | 1e-6    | 1e-5    | MLP     | 1e-6    | 1e-5    |
| CNN     | 1e-6    | 1e-5    | CNN     | 1e-6    | 1e-5    |
| VGG     | 1e-31   | 3e-5    | VGG     | 1e-30   | 3e-5    |
| WRN     | 1e-24   | 1e-5    | WRN     | 1e-24   | 1e-5    |
| WRN CA  | 1e-30   | 1e-5    | WRN CA  | 1e-25   | 1e-5    |

*Table 5.* Hyperparameters settings in WEISSI. Left: standard training, right: adversarial training.

**Baselines.** Three regularization methods are used for comparison baselines:

(i) *Weight Decay (WD)* takes the sum of L2 norm of all layers as the regularization term, as in (6).

(ii) *Parseval OC network* (Cisse et al., 2017) enforces that the rows of weight matrices are orthogonal, and its largest singular value is close to 1.

(iii) *Parseval network* (Cisse et al., 2017) imposes convexity constraints in residual aggregation layers (as in Eq. 39), in addition to the orthogonal constraints.

Note that in our experiment we update all rows of each weight matrix instead of a subset of rows as in the original paper (Cisse et al., 2017). For Parseval network, we use a parameterized function to ensure a valid convex aggregation, $a = \exp(l_a)/(\exp(l_a) + \exp(l_b))$ and $b = 1 - a$, instead of using the simplex projection.

**Training.** We minimize the softmax loss function using the momentum optimizer (Polyak, 1964) for all neural networks. For training on CIFAR10, we use data augmentation of random cropping and left/right flipping. Pixel values are normalized to $[0, 1]$. The batch size is set to be 100 for all experiments. Note that we do not use Batch Norm in our experiments because it makes the network invariant to the scale of weight norms due to the layerwise normalization, and it has been shown that Batch Norm hurts the adversarial robustness (Summers & Dinneen, 2020). We trained WideResnet with Batch Norm on CIFAR10 and it turns out although the accuracy is increased to 95.09%, it drops to 34.24% under FGSM attack when $\epsilon = 0.01$. For VGG network we use He initialization (He et al., 2015), while for WRN we use Fixup initialization (Zhang et al., 2019) to avoid gradient explosion (Zhang et al., 2019).

In addition to standard training, we also apply the adversarial training in (Madry et al., 2018), minimizing the loss:

$$\min_{\boldsymbol{\Theta}} \mathbb{E}_{(\boldsymbol{x}, \boldsymbol{y}) \sim \mathcal{D}} \max_{\boldsymbol{\delta} \in \mathcal{B}_{\Delta}} \mathcal{L}_{ce}(\boldsymbol{\Theta}, \boldsymbol{x} + \boldsymbol{\delta}, \boldsymbol{y}) + \lambda \mathcal{L}_{reg}(\boldsymbol{\Theta}),$$

where $\mathcal{B}_{\Delta}$ is the set of allowed perturbations. We choose the $l_{\infty}$-ball as the set of allowed perturbations and $\Delta = 0.01$ as the maximum $l_{\infty}$ distance.

**Evaluation.** We evaluate the generalization ability using the test accuracy on clean examples, and evaluate the adversarial robustness using the test accuracy on adversarial examples generated from two attacks, FGSM (Goodfellow et al., 2015) and PGD (Madry et al., 2018) with 10 steps (denoted as PGD10). For MNIST, we run 5 initializations and report the mean and standard deviation. For CIFAR10, we run 3 trials and choose the model with of median test accuracy.

The experiment is implemented in Tensorflow. For weight decay, we set the $\lambda_{wd} = 0.0001$. For Parseval network, we set the same $\lambda_{wd} = 0.0001$ for the weight decay of output layer and use $\beta_{parseval} = 0.0001$ in the orthogonal constraint updates. The hyperparameters in WEISSI are listed in Table 5. Notice that in some neural networks like VGG and WRN, we use different hyperparameters in adversarial and standard training. In MNIST experiment, MLP has two hidden layers with a width of 1024 and CNN has two convolution layers, where each layer has 128 filters of $3 \times 3$ size and the convolution stride is 1. We train the network for 60 epochs with an initial learning rate of 0.1. In CIFAR10 experiment, we use a variant of VGG16 for $32 \times 32$-sized input images, where the convolution layers are the same as VGG16 but we use one 512-width fully connected layer, and WideResnet-28-10. In VGG training, we train the network for 300 epochs with an initial learning rate of 0.01 and dropout rate of 0.5. The learning rate is decayed exponentially every 100 epchs with a decay rate 0.1. In WRN training, we set the training epoch as 200, initial learning rate as 0.03 and use the same learning rate decay scheme as in VGG.

**Ablation study.** We investigate the effect of $\mathcal{L}_{we}$ and $\mathcal{L}_{wc}$ on adversarial robustness of WRN by removing either term in the regularization. Fig. 4 shows a comparison between full WEISSI CA and only $\mathcal{L}_{we}$ or $\mathcal{L}_{wc}$ regularization when using FGSM and PGD attacks with different perturbation scales. WEISSI CA is robust to both attacks while single $\mathcal{L}_{we}$ or $\mathcal{L}_{wc}$ can only defend against one attack. Note that regularizing $\mathcal{L}_{we}$ benefits robustness to FGSM, while regularizing $\mathcal{L}_{wc}$

increases robustness to PGD. This suggests sparsity is crucial for defending against PGD and a smoother decision boundary is beneficial to defending against FGSM.

**Visualization.** We visualize the change of $\mathcal{L}_{we}$ and $\mathcal{L}_{wc}$ during WRN training under the 5 regularization schemes in Fig. 3. WEISSI achieves the lowest weight energy, while WEISSI CA shows a consistent decrease of weight complexity, indicating that the convex aggregation also helps train a more sparse network.

**Training with a Robust Loss Function.** Besides adversarial training, we test the performance of WEISSI using standard training with a robust loss function, Max-Mahalanobis center (MMC) loss (Pang et al., 2020). The regularization functions are the same as before. Table 4 shows a comparison between the performance of WEISSI and WD for VGG16 and WRN using standard training with MMC. In this case, WEISSI and WD achieve similar generalization performance, but WEISSI outperforms WD on the VGG network in terms of adversarial robustness.