# OpenReview forum: "Improve Generalization and Robustness of Neural Networks via Weight Scale Shifting Invariant Regularizations"
_ICML.cc/2021/Workshop/AML — ICML 2021 Workshop AML Poster_

### Official Review · Reviewer_QnVy · 2021-06-20
**a work with good insight into the ineffectiveness of weight decay regularizer and proposing a novel regularizer to regularize the intrinsic norms of  the network.**

**Rating:** Accept
**Confidence:** 4

**Review:**

This work is strict and well formulated in math, though some sentences are too long to follow.

The novelty of this paper is (1) discovering the ineffectiveness of weight decay regularizer (2) proposing a novel regularizer to regularize the intrinsic norms of  the network which is invariant to shifting of weight scales between layers.

Experiments are exhausted to demonstrate the effectiveness of WEISSI. It works for different attack methods and datasets.

The paper is too crowded: the margin below section 3. 4. 5. is too narrow, and there is no space for Conclusion section.

---

### Decision · Program_Chairs · 2021-06-21

**Decision:**

Accept (Poster)

**Comment:**

This paper proposes a novel regularizer to regularize the intrinsic norms of network. Experiments demonstrate the effectiveness for different attack methods and datasets.